# Widespread Reassortment Contributes to Antigenic Shift in Bluetongue Viruses from South Africa

**DOI:** 10.3390/v15071611

**Published:** 2023-07-23

**Authors:** Antoinette Van Schalkwyk, Peter Coetzee, Karen Ebersohn, Beate Von Teichman, Estelle Venter

**Affiliations:** 1Agricultural Research Council—Onderstepoort Veterinary Institute, Onderstepoort 0110, South Africa; 2Department of Veterinary Tropical Diseases, Faculty of Veterinary Science, University of Pretoria, Onderstepoort 0110, South Africa; dr.peter.coetzee.phd@gmail.com (P.C.); karen.ebersohn@up.ac.za (K.E.); estelle.venter@jcu.edu.au (E.V.); 3Design Biologix CC, Pretoria 0181, South Africa; beate@designbio.co.za; 4School of Public Health, Medical and Veterinary Sciences, Discipline Veterinary Science, James Cook University, Townsville 4811, Australia

**Keywords:** bluetongue virus, phylogenomic analysis, reassortment, antigenic shift, genetic diversity

## Abstract

Bluetongue (BT), a viral disease of ruminants, is endemic throughout South Africa, where outbreaks of different serotypes occur. The predominant serotypes can differ annually due to herd immunity provided by annual vaccinations using a live attenuated vaccine (LAV). This has led to both wild-type and vaccine strains co-circulating in the field, potentially leading to novel viral strains due to reassortment and recombination. Little is known about the molecular evolution of the virus in the field in South Africa. The purpose of this study was to investigate the genetic diversity of field strains of BTV in South Africa and to provide an initial assessment of the evolutionary processes shaping BTV genetic diversity in the field. Complete genomes of 35 field viruses belonging to 11 serotypes, collected from different regions of the country between 2011 and 2017, were sequenced. The sequences were phylogenetically analysed in relation to all the BTV sequences available from GenBank, including the LAVs and reference strains, resulting in the analyses and reassortment detection of 305 BTVs. Phylogenomic analysis indicated a geographical selection of the genome segments, irrespective of the serotype. Based on the initial assessment of the current genomic clades that circulate in South Africa, the selection for specific clades is prevalent in directing genome segment reassortment, which seems to exclude the vaccine strains and in multiple cases involves Segment-2 resulting in antigenic shift.

## 1. Introduction

Bluetongue (BT) is a non-contagious, *Culicoides*-spp.-transmitted viral disease of ruminants that is caused by the bluetongue virus (BTV), a species of the genus *Orbivirus* in the family *Reoviridae* [1,2]. The disease has a worldwide distribution in temperate and subtropical regions whilst constantly extending its distribution due to the influence of climate change, as well as anthropological factors, such as the increased international movement of animals and animal products [3,4,5,6]. In susceptible sheep, BTV causes BT, a disease that is characterised by fever, oedema, tissue infarction, coronitis, oral and nasal lesions, loss of condition, wool breaks, haemorrhage, reproductive difficulties, and variable mortality [7]. Bluetongue virus contains a segmented dsRNA genome consisting of ten linear genome segments ad approximately 19 kb in size. The genome encodes seven structural proteins (VP1-VP7) and five non-structural proteins (NS1 to NS4) that show varying levels of sequence conservation [8,9]. The highly variable outer capsid protein 2 (VP2) is the main serotype-determining antigen [10], and 28 serotypes of the virus have been identified to date [11], with a few further putative serotypes identified in Europe, the Middle East, and Asia [12,13,14]. Bluetongue is a notifiable disease for the World Organisation for Animal Health (WOAH) due to its economic importance, which includes both direct and indirect losses [15].

The virus is endemic throughout South Africa, with 21 of the 28 known serotypes having been detected. It has a seasonal prevalence coinciding with increased rainfall. Outbreaks start in September, and peak infection numbers are reached from February to April [16]. Different serotypes are annually associated with outbreaks, with the most prevalent identified as BTV-1, BTV-2, BTV-3, and BTV-4, dominating in certain years, just to be replaced with others the following year [16]. At least 13 livestock-associated *Culicoides* spp. are implicated vectors of BTV in South Africa. BTV has been isolated from five of these species. This implies that a multi-vector potential for the transmission of BTV exists, which complicates the epidemiology of BT in South Africa [17]. The two *Culicoides* spp. recognised as the main vectors of BTV in South Africa are *C. imicola*, responsible for the transmission of BTV throughout most of South Africa, and *C. bolitinos*, which plays a role in the transmission of the virus in the cooler, higher-lying regions of the country [17].

South Africa has a sheep population of nearly 21.6 million that is distributed unevenly throughout the country’s nine provinces [18]. Although sheep are recognised as the predominantly affected species, cattle and goats, which are largely sub-clinically affected, are thought to act as reservoir hosts of the virus. The epidemiological situation in South Africa is further complicated by the presence of a large population of unvaccinated, clinically resistant indigenous sheep breeds and wild ruminant species that play an undefined role in propagating the disease [19]. 

South Africa was the first country to successfully develop a live attenuated vaccine (LAV) against BT, which is still used in the country today [20]. This vaccine consists of 15 serotypes (BTV-1 to BTV-14, and BTV-19) that were attenuated via serial passage in embryonated chicken eggs and cell culture [21]. The use of the vaccine in the field has led to a situation where both wild-type and vaccine strains co-circulate, potentially leading to reassortment and recombination and the potential emergence of new viral strains with novel phenotypic characteristics [22,23,24,25]. This is particularly noteworthy, as the live attenuated vaccine strains have demonstrated novel phenotypic properties such as increased transmissibility by *Culicoides* and an increased capability to cross the ruminant placenta, thereby leading to teratogenic defects [22].

The identification of the serotype of circulating viruses is important from an epidemiological perspective and for vaccine antigen selection since the viral serotypes show varying levels of cross-neutralisation [21]. Since BTV contains a segmented genome, the reassortment of segments both within and between serotypes can occur [26]. The relative role of reassortment in the evolution of the virus has been investigated for BTV isolates from America, Europe, and India, which has indicated that reassortment occurs regularly between the segments of different co-circulating viral strains and serotypes [24,27,28,29]. An investigation of the frequency of genetic reassortment indicated that there was a bias in the segments that were reassorted and that certain genome segments tended to be reassort together [6,29]. Additionally, genomic sequence analysis indicated that reassortment was followed by a high frequency of amino acid changes in the reassorted genome segments, indicative of possible adaptive evolutionary changes in the reassorted viruses [24]. Besides reassortment, BTV has also been demonstrated to evolve through genetic drift and genetic recombination [30,31]. Cumulatively, these processes result in high plasticity in the genome of BTV, which is thought to be the main driver for the emergence of strains that demonstrate different phenotypic properties [22,24,25,30].

Little is known about the molecular epidemiology or evolution of the virus in the field in South Africa [32]. Only a single study has been performed, in the 1990s, in which the molecular epidemiology of the virus was described, based on sequencing of the NS3 gene of a limited number of isolates [33]. This study failed to address the relative role of genetic drift, shift, and genetic recombination in the evolution of the virus in South Africa. In recent years, next-generation sequencing (NGS) of all the viral genome segments has become the gold standard for investigating the evolution and spread of orbiviruses. In the case of BTV, NGS has enabled investigation into the role that genome segment reassortment, genetic recombination, and genetic drift have played in the evolution of the virus [24,25,27,28,29]. Additionally, this has allowed topotypes of the different genome segments to be identified [33], allowing for the tracking of the “flow” of different genetic versions of the genome segments in regions of endemicity [34], and has allowed for the delineation of the global phylogeographic history of the virus [35]. The analysis of the whole genome of BTV has also in some cases allowed for the identification of the potential unauthorised use of live attenuated BTV vaccines in BTV-non-endemic regions [24,25].

The purpose of this study was to contribute to the knowledge of the genetic diversity of field strains of BTV in South Africa (genotypes, serotypes and topotypes) and to provide an initial assessment of the evolutionary processes shaping BTV genetic diversity in the field. To do this, the genomes of 35 field viruses belonging to 11 serotypes that were collected from different regions of the country between 2011 to 2017 were sequenced. A phylogenetic analysis allowed for an initial assessment of the current genomic clades that circulated in South Africa and allowed for the detection of genome segment reassortment events between wild-type and vaccine strains.

## 2. Materials and Methods

### 2.1. Cell Lines, qRT-PCRs, and Viruses

Blood samples were collected from sheep demonstrating clinical signs of BT, in different regions of South Africa, during the 2011–2017 vector seasons. The heparin blood samples were submitted to the Veterinary Virology Laboratory, DVTD, Faculty of Veterinary Science, University of Pretoria, for virus isolation. This was performed in baby hamster kidney cells (BHK), grown to confluence using Eagles Minimum Essential Medium (EMEM) (Gibco, Life Technologies, Waltham, MA, USA) supplemented with 10% tryptose phosphate broth (*v*/*v*) and 5% foetal bovine serum (FBS) (*v*/*v*) (Gibco, Life Technologies, Waltham, MA, USA). 

The presence of BTV in CPE-positive cell cultures was verified through RT-qPCR using BTV primers and a Segment-1-specific hydrolysis probe [36]. Samples that demonstrated a Ct of <34 were processed for viral dsRNA extraction. The live attenuated BTV vaccine (Bottles A, B and C) was procured from OBP in Pretoria, South Africa. Virus isolation, serotyping, and complete genome sequencing of Bottles A and C have been described previously [37,38]. In this study, Bottle B was processed according to the same methods as previously described. The serotype of vaccine plaques was determined using the SuperScript™ III One-Step RT-PCR System with Platinum™ Taq DNA Polymerase, according to the manufacturer’s instructions, with 10 µM each of serotype-specific forward and reverse primers, as previously published [37]. Serotype identities were determined by analysing amplicon sizes using agarose gel electrophoresis, or by sequencing the amplicons using either the forward or reverse primer, as appropriate.

### 2.2. Purification of dsRNA, Full-Length Amplification of Viral cDNAs (FLAC), and Illumina Sequencing

Field and plaque-purified virus isolates were passaged on confluent BHK cells, and total RNA was extracted using TRIzol (Invitrogen, Life Technologies, MA, USA) according to the manufacturer’s instructions. The total RNA was dissolved in 90 µL TE buffer (pH, 7.4), to which 30 µL of 8 M LiCl was added. Double-stranded RNA was purified using LiCl precipitation, and the quality and integrity of the dsRNA were evaluated using 1% TAE agarose gel electrophoresis as previously described [38].

Sequencing templates were prepared using sequence-independent whole-genome reverse transcription-PCR (RT-PCR) amplification [39]. Polymerase chain reaction amplicons were sequenced at 7500× coverage on an Illumina MiSeq sequencer (Inqaba Biotechnical Industries [Pty] Ltd., Pretoria, South Africa). The Nextera XT DNA sample preparation kit and 300-bp paired-end V3 Illumina chemistry were used to conduct the sequencing. Illumina sequence reads were analysed using Geneious 8.1.5, and a combination of de novo assembly and read mapping to reference strains of the same serotype was used to obtain the full-length genome sequences of the viruses. The reads were mapped to the consensus sequence in order to detect any variants and subsequently determine that a single virus was presented in the reads. The origins, passage histories, and Genbank accession numbers of the BTV isolates used in this study are listed in Appendix A.

### 2.3. Sequence Analysis

These genomes were analysed using maximum likelihood phylogenetic analysis and were subjected to reassortment and recombination detection using the Cluster Picker and the Recombination Detection Program version 4 (RDP4), respectively. Included in the analysis were the full genome sequences of 15 vaccine strains that are currently included in the commercial Onderstepoort Biological Products vaccine, the serotypes 1 to 24 of the original “Howell” prototype reference strains obtained from the Department of Veterinary Tropical Diseases, University of Pretoria (DVTD-UP) collection, as well as the 24 reference strains available from the WOAH Reference Centre at the Agricultural Research Council—Onderstepoort Veterinary Institute (ARC-OVI) [39,40,41]. Individual sequence alignments of each of the ten genome segments, containing the full-length sequence of 35 field viruses, 15 South African commercially available vaccine strains, 24 ARC-OVI-Reference collection, and 24 isolates from the DVTD-UP–“Howell” collection, as well as 209 complete BTV genome sequences from GenBank, representing isolates from Africa, the Middle East, Europe, Asia, the Americas, and Australia (total n = 305), were generated using CLC Genomics Workbench v.9.5 (Qiagen, www.clcbio.com (accessed on 30 November 2016). The alignments were individually used to determine the phylogenetic relatedness of the viruses by generating individual maximum likelihood trees in Mega X [42]. Each phylogenetic tree was constructed under General Time Reversal (GTR) (G+I, G = 4) with 1000 bootstrap iterations. Both the original alignment and newly generated phylogenetic trees of each segment were used in Cluster Picker to identify monophyletic clades based on a distance threshold for each cluster of 5% [43]. The time to the most recent common ancestor (TMRCA), the substitution rate, and the selection were determined as previously described [44]. 

## 3. Results

Viruses were successfully isolated, representing 11 different serotypes (BTV-1, BTV-2, BTV-3, BTV-4, BTV-5, BTV-7, BTV-12, BTV-13, BTV-16, BTV-17, and BTV-24) and spanning five years of outbreaks (2011, 2014–2017) (Figure 1). Multiple serotypes were identified in a single year from a single province, as well as a single serotype in multiple provinces during the same season (Figure 1). The complete genome sequences of the field isolates were determined and compared to the 24 viral sequences representing the first 24 serotypes from the ARC-OVI-Reference and the DVTD-UP–“Howell” collections, the 15 LAVs in the commercially available vaccine, and 209 isolates from Africa, Europe, Asia, the Middle East, the Americas, and Australia. Individual sequence alignments were generated for each of the ten genome segments, and these were used in subsequent analysis. The phylogenetic relationships of the 305 viruses were determined with maximum likelihood phylogenetic trees, and in turn, these trees were used to cluster the sequences based on a genetic distance threshold of 5% per cluster (Figure 2). 

Sequences from all 305 sequences per segment were clustered using a genetic distance threshold of 5%. Segment-1 was divided into 36 clusters, and the South African (RSA) isolates belonged to 10 of them (Figure 2). Segment-3 was divided into 40 (RSA = 4), Segment-4 into 49 (RSA = 7), Segment-5 into 39 (RSA = 5), Segment-6 into 49 (RSA = 12), Segment-7 into 39 (RSA = 13), Segment-8 into 50 (RSA = 12), Segment-9 into 42 (RSA = 10), and Segment-10 into 47 clusters (RSA = 16) (Figure 2). Clusters containing an ARC-OVI reference sequence were named according to the corresponding serotype or serotypes (Figure 2), whilst clusters containing new field isolates were labelled alphabetically based on the segment number. For example, Segment-1 had clusters A1 to A7, and Segment-3 had clusters B1 to B3 (Figure 2). This nomenclature would be used in subsequent descriptions of the individual segments. 

The phylogenetic analysis of Segment-2 (VP2), responsible for clustering the viruses based on their serotype, indicated a close association of the viruses from South Africa with those isolated in Africa, Europe, and the Middle East (Figure 3). BTV-1_VR27_RSA_2017 shared 100% sequence identity of all 10 segments with the reference isolate BTV-1_Biggardsberg_KZN_RSA_1958, as well as the BTV-1 LAV, available in the commercial vaccine. It can therefore be assumed that this 2017 isolate originates from the LAV. The remaining Serotype 1 viruses had a close phylogenetic clustering with isolates from Africa and Europe, an observation that was mirrored in Serotypes 2, 3, 4, 5, 13, and 24 (Figure 3). In contrast, the two Serotype 5 RSA isolates clustered either with the reference isolates or with the isolates from India in 2015, whilst the RSA isolates from Serotype 17 clustered with each other in a unique lineage (Figure 3). 

It was observed that the RSA isolates did not follow the same phylogenetic clustering across all ten genome segments as described for Segment-2 but rather that Segment-2 was the reassorted segment in the genetic backbone of other co-circulating segments. These will be discussed in more detail. 

Segment-1 had a phylogenetic clustering separating the isolates from Africa, Europe, and the Middle East into three larger groups, whilst the samples from the Americas and samples from Australia and India formed two separate groups (Figure 4A). The RSA isolates predominantly displayed a geographical clustering of this segment, with samples from the same or adjacent provinces grouping within the same sub-clusters. Subgroup 1 contains five sub-clusters, with samples from the Eastern Cape (EC) and one from the Free State (FS) provinces clustering with the reference sequence of Serotype 22 (Figure 4B). Similarly, the sequence (BTV-3_VR03_RSA_2014) obtained from the Northern Cape (NC) Province clustered with Serotype 24 but was not identical to the reference isolate (Figure 4B). This indicates genetic drift from samples that were circulating in South Africa in the 1960s, resulting in the field isolates observed in 2017 (Figure 4B). Three additional sub-clusters were identified in Subgroup 1. Whilst two of the isolates from the Western Cape (WC) Province clustered with isolates from Africa and Europe (Cluster A2 and A5) (Figure 4B), the isolate from the North West (NW) Province formed a monophyletic lineage (Cluster A6) (Figure 2 and Figure 4B). Subgroup 2 consists of four sub-clusters, predominantly containing samples from the FS and Mpumalanga (MP), as well as samples from Gauteng (GP), KwaZulu-Natal (KZN), and the EC (Figure 4C). None of these four sub-clusters were associated with a reference strain but did include samples from Africa, the Middle East, and Europe, except Cluster A3, which consisted only of RSA field isolates (Figure 4C). The genomic segments of isolate BTV-4_VR01_RSA_2017, obtained from Rosendal in the FS, grouped with samples from the EC concerning Segment-1, -3, -4, -5, -7, -8, -9, and -10, rather than samples from the FS. A similar clustering pattern described for the RSA isolates in Segment-1 was observed for other segments and will be individually described below.

When all 305 sequences of Segment-3 were analysed, the percentage sequence identity was significantly high, with all the isolates belonging to only four major clusters (Figure 5A). These clusters consist of subgroup 1, containing all the RSA isolates, as well as samples from Africa, the Middle East, and Europe; a sub-cluster consisting of samples from the Americas; one containing samples from Australia and India; and the last sub-cluster containing the reference strains of BTV-1 to BTV-5, BTV-9 to BTV-12, BTV-14, BTV-15, and BTV-19 (Figure 5A). Subgroup 1 consisted primarily of sub-cluster B1, associated with reference strains BTV-18, BTV-22, and BTV-24 and containing 32 of the RSA isolates (Figure 2 and Figure 5B). Similar to the sequences of Segment-1, none of the 32 isolated sequences were identical to the reference sequences, but they shared a recent common ancestor with them (Figure 5B). An isolate from the FS (BTV-13_VR38_RSA_2017) formed a unique monophyletic sub-cluster (B2), whilst isolate BTV-7_VR43_RSA_2017 from the EC formed a smaller sub-cluster (B3) with a previous isolate from RSA (Figure 5C).

Analyses of Segment-4 indicated the isolates from RSA, Africa, the Middle East, and Europe cluster into a single major group, Group 1, consisting of seven sub-clusters (Figure 6A). Two of the sub-clusters are associated, respectively with BTV-22 and BTV-24, suggesting genetic drift from a recent common ancestor (Figure 6). Samples originating from the FS and MP provinces clustered into four sub-clusters each, whilst samples from the EC were divided into two sub-clusters (Figure 6). Three samples from the EC (BTV-1_VR06_RSA_2017, BTV-1_VR13_RSA_2017, and BTV-17_VR17_RSA_2017), as well as sample BTV-4_VR01_RSA_2017 from Rosendal in the FS, which clusters together in Segment-1, -3, -5, -7, -8, and -9, were subdivided into clusters C3 and C1, indicating possible reassortment of Segment-4 in sample BTV-1_VR13_RSA_2017 (Figure 6A). These four isolates had another possible reassortment event in Segment-10 when the sample BTV-17_VR17_RSA_2017 clusters with the reference strains BTV-1, 6, 9 and 13, whilst the other three isolates form Cluster I3. Additionally, these four isolates were examples of possible reassortment events involving the antigenic determinants encoded by Segment-2 and Segment-6, with two isolates belonging to BTV-1, one to BTV-4, and one to BTV-17 (Figure 3 and Figure 11). 

The analysis of Segment-5 identified two groups containing samples from South Africa, Africa, Europe, and the Middle East, which were distinct from the samples that originated in the Americas, Australia, and India (Figure 7A). The RSA isolates were clustered into five smaller groups, with four of them associated with reference strains BTV-1, -6, -2, -18, and -24 (Figure 7). The relationships to the reference strains were not significant to assume that the segments were recent acquisitions from LAV strains, except for the previously described sample BTV-1_VR27_RSA_2017 (Figure 7). Similar to Segment-4, samples from the FS clustered into four subgroups, while the samples from the EC were grouped into two sub-clusters (Figure 7). The separation of the EC samples is the first and only subdivision of the four BTV-12 isolates, with two isolates (BTV-12_VR54_RSA_2017 and BTV-12_VR55_RSA_2017) belonging to Cluster-B2 and two isolates (BTV-12_VR44_RSA_2017 and BTV-12_VR16_RSA_2017) in Cluster D1 (Figure 7B). This is the only example of reassortment between these four BTV-12 isolates and one of the other EC samples that belongs to the same cluster, Cluster D1. The origin of this segment in the EC Province is not clear. Similarly, isolate BTV-4_VR31_RSA_2017 in Cluster BTV-18 seems to be another possible reassortment, since it groups together with the other BTV-4 isolates (BTV-4_VR42_RSA_2017, BTV-4_VR34_RSA_2017, and BTV-4_VR48_RSA_2017 in Cluster D1, Figure 2) according to all ten segments except Segment-5 and -8 (Figure 7 and Figure 8). 

The second NS2 protein, encoded by Segment-8, has three clusters associated with reference sequences (Figure 8). Yet, similarly to previously described RSA sequences clustering with reference isolates, the sequences in clusters BTV-18 (BTV-1_VR13_RSA_2017, BTV-1_VR06_RSA_2017, BTV-4_VR01_RSA_2017, and BTV-17_VR17_RSA_2017) and BTV-22 (BTV-12_VR16_RSA_2017, BTV-12_VR44_RSA_2017, BTV-12_VR54_RSA_2017, and BTV-12_VR55_RSA_2017) share a recent common ancestor with the reference isolate but are not identical to the reference strains (Figure 8). The remaining RSA sequences were clustered into eight groups of isolates from Africa, the Middle East, and Europe. The RSA isolates had an association based predominantly on the geographical provinces they were isolated from, with the previously described isolates from the FS clustering predominantly with samples from MP Province (Figure 8). The possible reassortment concerning the BTV-4 isolates has been mentioned, but additionally, another possible reassortment is described in isolate BTV-16_VR08_RSA_2017 (Cluster G5), which predominantly clusters with isolates BTV-3_VR22_RSA_2016, BTV-3_VR11_RSA_2017, and BTV-3_VR33_RSA_2017 (Cluster G1). The latter isolates cluster in the same groups for Segment-1, -3, -4, -5 and -6, whilst they were divided into two clusters for Segment-2, -8, and -9 and three clusters for Segment-7 and -10.

Segment-9 sequences from RSA were all clustered into a single major group, called Group 1 (Figure 9A). This clustering of sequences from RSA, Africa, the Middle East, and Europe resembles the same phylogenetic pattern as observed for sequences from Segment-3. The isolates from RSA were subdivided into ten clusters, two of which were associated with the reference strains BTV-1 and BTV-24 (Figure 2). In contrast to sample BTV-1_VR27_RSA_2017, which is also identical to the reference strain and probably originated from the LAV (Figure 9C), isolate BTV-2_VR18_RSA_2017 is related to the reference strain of BTV-24 (Figure 9B). Three isolates from the WC Province (BTV-2_VR62_RSA_2017, BTV-4_Ewe-S74_RSA_2017, and BTV-24_VR25_RSA_2017) clustered together in Segment-3 and -9, whilst in Segment-1 and -4, different combinations of these isolates clustered together. All of the isolates from the EC Province, except BTV-2_VR18_RSA_2017 (Cluster BTV-24), grouped into a single cluster (H1), indicating a bias towards this sequence in the region (Figure 9B).

In contrast to the distinct separation observed in Segment-1, -3, -4, -5, -8, and -9 between isolates from Africa, the Middle East, and Europe compared to the Americas, Australia, and India, the remaining segments did not display the same segregation. Sequence analysis of Segment-7, encoding the major core protein VP-7, was divided into four groups (Figure 10A). The major group, Group 1, contained samples from Africa, the Middle East, Europe, and the Americas (Figure 10B) and included 18 of the RSA isolates in a single cluster, BTV-1/18. There were five additional RSA samples in Group 1, but they represented two smaller clusters (Figure 10B). The smaller group, Group 2, contained samples from RSA clustered into two sub-clusters, as well as isolates from Africa, Europe, India, and Australia (Figure 10D). Samples from these continents were included in Group 3, as well as eight RSA samples divided into seven sub-clusters (Figure 10C). Group 4 forms a distant clade to the remainder of the sequences of Segment-7 (Figure 10A). This group consists primarily of samples of serotypes BTV-7, -13, -15 and -19 from Africa, Australia, and the USA, as well as a single RSA sample (BTV-7_VR43_RSA_2017) that forms a monophyletic lineage (Cluster F8) (Figure 10E).

Segment-6 encodes the second capsid protein VP5 and predominantly follows a close association with Segment-2. The sequences were clustered into three major groups and a small group consisting of isolates from serotypes BTV-7 and BTV-19 (Figure 11A). Similarly to the observations of Segment-7, isolate (BTV-7_VR43_RSA_2017) clusters within the BTV-7 and BTV-19 sub-cluster, which forms a unique lineage (Figure 11E). As expected, the isolates cluster according to serotypes, with Group 1 containing the RSA isolates of BTV-3, -4, -5, -13, -17, and -24 (Figure 11B). Group 2 consists of serotypes BTV-1 and 02, while Group 3 contains the isolate of serotype BTV-12 (Figure 11C,D). 

Segment-10 encodes the non-structural protein (NS3) and is genetically the second most variable, after Segment-2. The RSA isolates were grouped into 16 different clusters, with 6 of these containing a single sequence (Clusters I6, I7, I8, I12, I15, and I20; Figure 2). Similar to Segment-6 and -7, isolate BTV-7_VR43_RSA_2017 clusters in a unique lineage associated with the reference sequences of BTV-19 (Figure 12A). The remaining RSA isolates were divided into either of the major groups (Group 1 or Group 2) along with sequences from Africa, the Middle East, Europe, the Americas, and Australia (Figure 12B,C). 

## 4. Discussion

It is predicted that multiple BTV serotypes circulate each vector season in South Africa, but since no ongoing surveillance for BT is performed in the country, the only data available on the occurrence of the virus are largely based on infrequently submitted diagnostic samples [45,46] (personal communication with the head of Virology section, ARC-OVI, 2023). This is the first study in recent years to investigate the serotypes involved during active BT outbreaks in South Africa and, more importantly, the first to determine the complete sequences of the BTVs isolated from these outbreaks. Unfortunately, this study is limited to outbreaks in 2011 and 2014–2017 and only incorporates a small number, less than 20% of the reported cases, circulating in the country during the same period [18]. Despite annual outbreaks of BT in South Africa, the reporting and funding remain extremely limited. The 35 BTV isolates represented 11 serotypes, with one isolate, BTV-1_VR27_RSA_2017, sharing a significantly high percentage sequence identity over the complete genome to the LAV Serotype 1 vaccine strain. The presence of LAV-derived sequences in the field virus population was anticipated, since annual vaccination using the only commercially available vaccine, based on the LAV of serotypes 1 to 14 and 19, is recommended in the country [18,21,46]. Except for the previously described isolate (BTV-1_VR27_RSA_2017), none of the other sequences generated from the new field viruses had a segment with a similarly high percentage sequence identity with any of the other LAV strains. Various segments shared sequence conservation, with a percentage sequence identity higher than the delineated 5% threshold with reference and LAV strains, but none of the field virus sequences were identical to the reference or LAV sequences. This implies that the new sequences were due to genetic drift from a common recent ancestor with the reference or vaccine strains, rather than recent reassortment with LAVs. This contrasts with previous studies from America and Europe, where repeated reassortment between field and LAV strains was described [25,28,47]. This could imply that, in contrast to regions where the presence of BTV is due to novel incursions, the highly diverse BTV population circulating in South Africa gains no phenotypical benefits from acquiring segments derived from LAVs. Alternatively, this could be due to the small number of isolates sequenced, which might not be representative of the natural population of BTV circulating in the country [18].

Based on the genetic clustering of the individual segments, it is evident that targeted selection for specific sequences occurs within a geographical region. This is despite the isolates belonging to different serotypes, suggesting an antigenic shift with the reassortment of Segment-2, the major antigen determinant. Examples of these include BTV-3_VR05_RSA_2017 and BTV-5_VR45_RSA_2017, which share six homologous segments (1, 3, 5, 6, 8 and 9), whilst isolates BTV-4_VR01_2017 from the FS and BTV-17_VR17_RSA_2017 from the EC provinces had seven homologous segments (1, 3, 4, 5, 7, 8, and 9). This increased selection for a specific variant has been reported previously, in which a particular Segment-5 sequence had an increased frequency in BTVs isolated from India [48]. The antigenic shift resulting from the reassortment of Segment-2 may provide novel phenotypic advantages to the reassortant viruses, such as the evasion of susceptible animals’ immune systems [49]. 

Since a bias towards sequences belonging to a limited number of clusters was observed, for example, the 10 segments of the 35 RSA isolates were grouped into only 100 clusters (Figure 2), additional investigation into the possible divergence time of these selected clusters was performed. The time to the most recent common ancestor (TMRCA) of these selected clusters was less than 120 years ago, and the majority of them (n = 69) are not associated with the reference strains. The latter essentially represent BTVs isolated during outbreaks in South Africa in the middle of the 20th century [39]. The ARC-OVI reference strains were used during the generation of the cell-culture-adapted LAV strains for the majority of the serotypes, resulting in the close phylogenetic clustering of reference and LAV strains [50,51]. This re-emphasises the observation that the sequences associated with the 39 reference clusters are related to the reference sequences but not identical and thus not obtained from reassortment with the LAVs. Regardless of the previously described selection for specific sequences, the recent field isolates are genetically highly variable, either through their constitution of different combinations of selected segments or through genetic drift within the segments. Only two isolates (BTV-12_VR54_RSA_2017 and BTV-12_VR55_RSA_2017) were identical, whilst four other isolates share the same genomic clustering pattern, with eight segments clustering together amongst the four isolates and the remaining two segments clustering together in three of the isolates (BTV-4_VR03_RSA_2017, BTV-4_VR34_RSA_2017, BTV-4_VR42_RSA_2017, and BTV-4_VR48_RSA_2017). Our data indeed support the concept of genetic “mixing” between multiple co-circulating strains, as previously suggested, and that reassortment is a non-random selection process selecting from a specific set of genomic segments [24,29].

The sequence homology across all the field isolates produced a strong temporal clustering, yet the majority of these sequences were not identical but rather shared a significantly high percentage sequence identity. Based on these observations, the role of selective pressure and substitution rates on the divergence from a common recent ancestor across serotypes and segments was investigated. Unsurprisingly, the segments with the lowest number of genetically distinct clusters (1, 3, 4, 5, 8, and 9) had similar substitution rates (~3.6 × 10^−4^ substitutions per site per year), TMRCA (~300 years ago), and all were predicted to be under purifying selection. The substitution rates and selection pressure were consistent with similar reports from Australia, Europe, and Asia [30,35,48,52]. These results support previous findings that the individual segments evolve independently under genetic drift [44]. Even though Segment-2 had a TMRCA of ~9000 years ago, the partitioning of individual serotypes predicted common ancestors around 300 years ago. Serotypes 16, 20, 21, 22, and 23 are either exotic or were not previously observed in South Africa and thus were removed from the molecular clock analysis [46]. Sequences belonging to these exotic serotypes were also identified as outliers during the root-to-tip regression plots of genetic distance against sampling time. Even though serotypes 7, 12, 15, and 19 were observed in South Africa in both the 20th and 21st centuries, sequences from Segment-2 and -6 belonging to these serotypes were outgroups, responsible for the increased TMRCA (Figure 3). Segment-2 of Serotype 12 had less than 44% sequence identity to all other serotypes except for Serotype 15 at 52% and Serotype 22 at 70%, indicating a shared evolutionary relationship outside of the other serotypes (Figure 3A).

The data presented in this study suggest that rather than diversifying selection pressures and genetic drift, the reassortment of preferred segments plays an important role in BTV evolution and is responsible for the increase in virus diversity. There are continuous calls for segmented virus nomenclature to reflect the diversity of all the segments, not only based on serotype or the genotype of a specific segment [24]. Based on the high genetic diversity of the field isolates analysed in this study, the current nomenclature system of virus isolation based on serotype is sufficient for BTV isolates from South Africa. This is due to the constant formation of new field-related clusters and the complete mixing of the genetic segments between viable genetic clusters. Based on the current clustering of field isolates, there are 3.5 × 10^6^ different possible ten-segment combinations.

The epidemiology and evolution of BTV in South Africa are significantly under-investigated and under-reported. The problem is exaggerated by annual outbreaks occurring in a BTV-endemic country where every vector season is preceded by vaccinating animals with 15 LAV vaccine strains. The role of *Culicoides* insects in the evolutionary dynamics of BTV in South Africa has also not been investigated, but it is hypothesised that it would contribute significantly to viral fitness and diversity. There is vast scope to investigate the natural population of BTV in an endemic region, to enhance our current understanding of the mechanisms involved in fitness or phenotype-driven evolution, and to subsequently implement that knowledge to improve prevention and global control strategies for the disease.

## Figures and Tables

**Figure 1 viruses-15-01611-f001:**
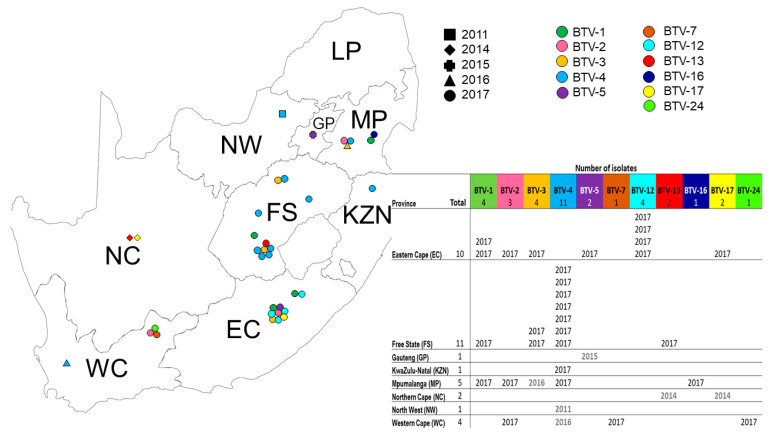
Map of South Africa indicating the locations of BTV isolates and the year of isolation according to the key provided. A summary of all the serotypes per province per year is provided in Appendix A.

**Figure 2 viruses-15-01611-f002:**
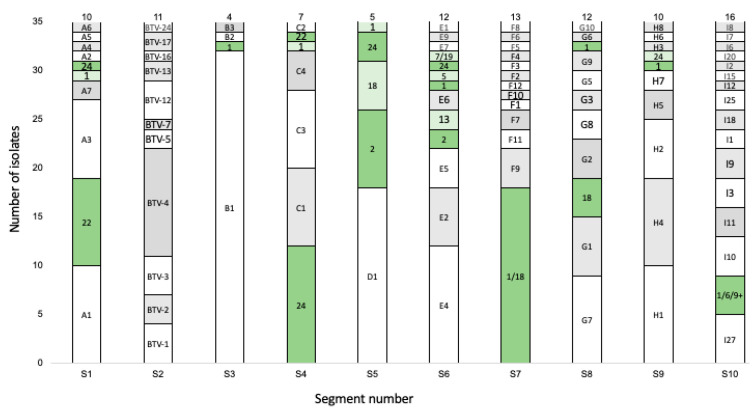
Graphical representation of the number of RSA samples divided into individual genetic clusters. The names of the clusters are indicated in the figure, whilst the total number of clusters are indicated at the top of the bar. Clusters containing an ARC-OVI reference sequence were given the name of the corresponding serotype or serotypes and are represented in green. The remainder of the clusters were named alphabetically, starting with Segment-1.

**Figure 3 viruses-15-01611-f003:**
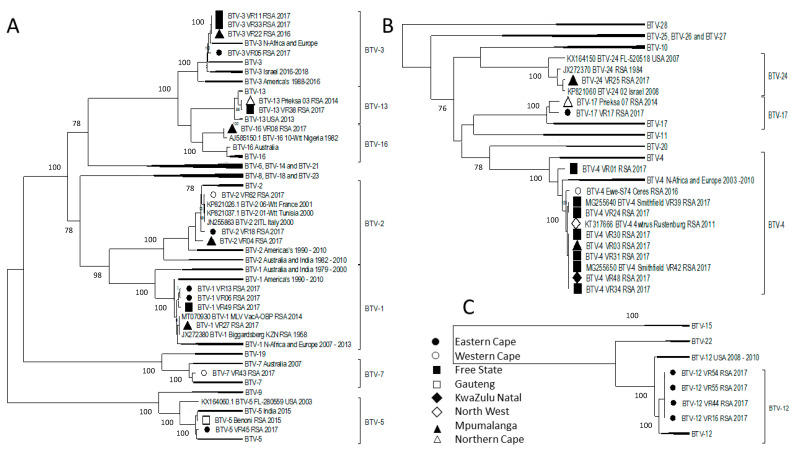
Phylogenetic analysis of Segment-2 sequences using maximum likelihood analysis. The phylogenetic tree was divided into three subtrees (**A**–**C**), and isolates from RSA are indicated based on the province they originated from (key provided in the figure).

**Figure 4 viruses-15-01611-f004:**
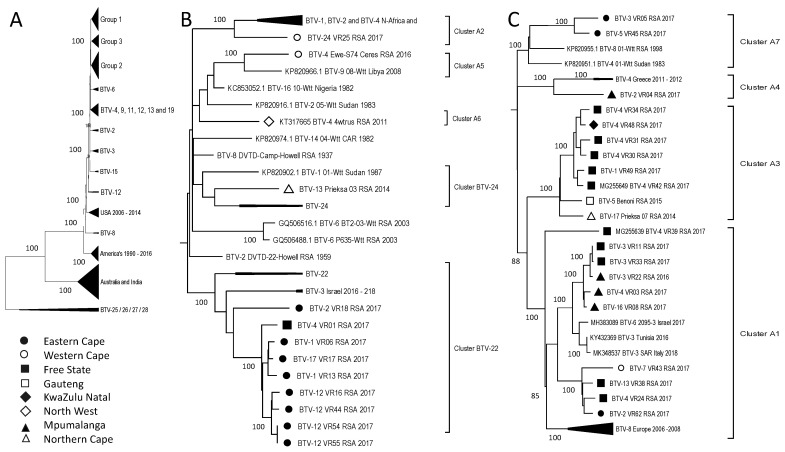
Phylogenetic analysis of Segment-1 sequences using maximum likelihood analysis. (**A**) A phylogenetic tree containing all the samples grouped into large clusters. (**B**) Subtree of group 1. (**C**) Subtree of group 2. Isolates from RSA are indicated based on the province they originated from (key provided in the figure).

**Figure 5 viruses-15-01611-f005:**
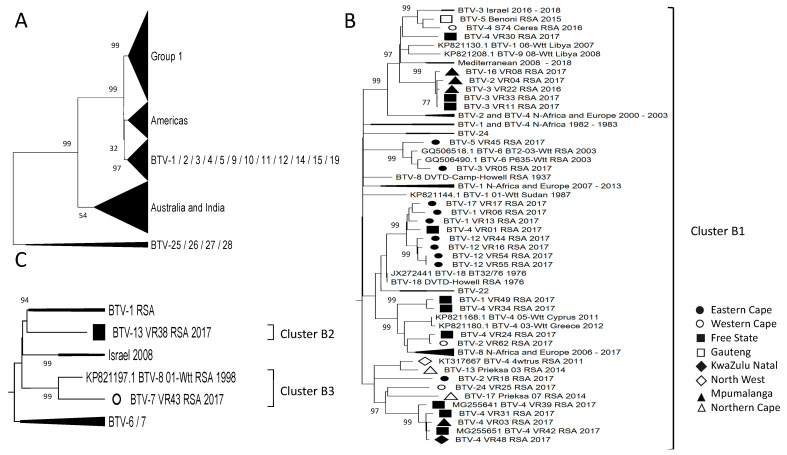
Phylogenetic analysis of Segment-3 sequences using maximum likelihood analysis. (**A**) A phylogenetic tree containing all the samples grouped into large clusters. (**B**) Major subtree from Group 1, containing Cluster B1. (**C**) Minor subtree within Group 1, containing clusters B2 and B3. Isolates from RSA are indicated based on the province they originated from (key provided in the figure).

**Figure 6 viruses-15-01611-f006:**
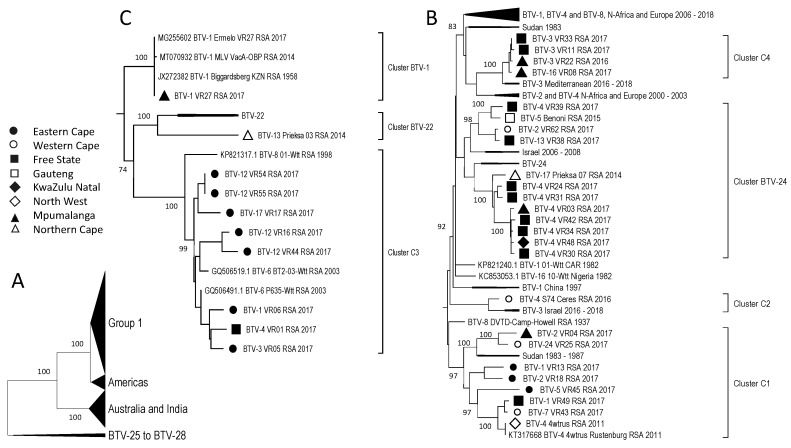
Phylogenetic analysis of Segment-4 sequences using maximum likelihood analysis. (**A**) A sub-phylogenetic tree containing a selection of the samples grouped into group 1. (**B**,**C**) The second subtree contains samples from group 1. Isolates from RSA are indicated based on the province they originated from (key provided in the figure).

**Figure 7 viruses-15-01611-f007:**
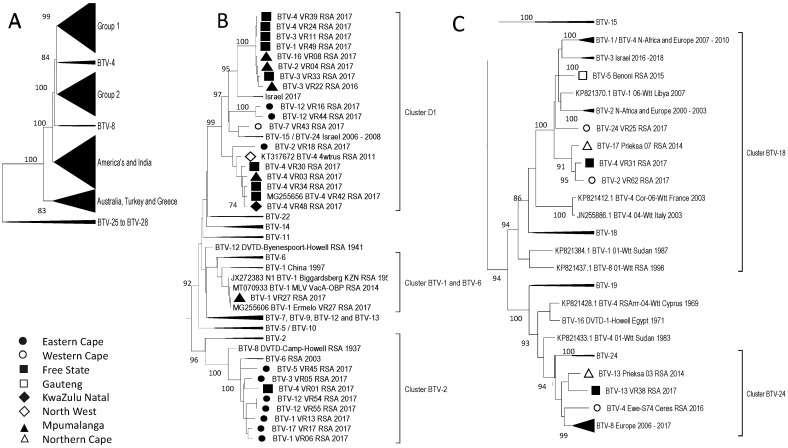
Phylogenetic analysis of Segment-5 sequences using maximum likelihood analysis. (**A**) A phylogenetic tree containing all the sequences clustering the RSA samples into two large groups, 1 and 2. (**B**) Subtree representing Group 1. (**C**) Subtree of isolates in Group 2. Isolates from the RSA are indicated based on the province they originated from (key provided in the figure).

**Figure 8 viruses-15-01611-f008:**
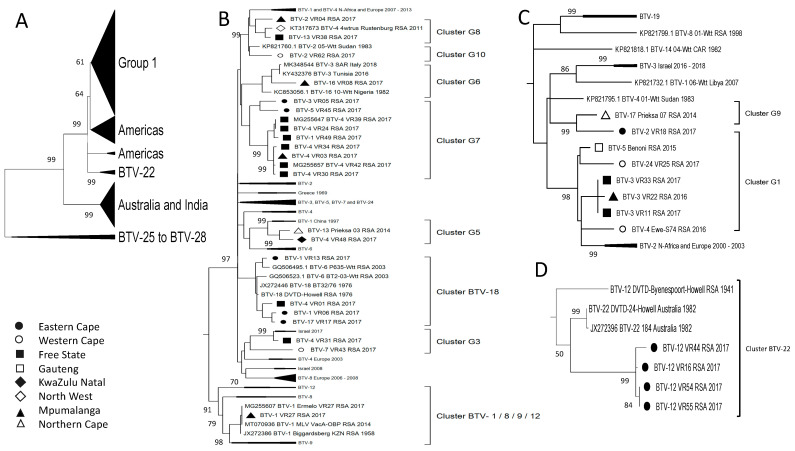
Phylogenetic analysis of Segment-8 sequences using maximum likelihood analysis. Samples from RSA belonged to either Group 1 or Cluster BTV-22. (**A**) A subtree from Group 1 containing the majority of the RSA samples. (**B**) The second subtree of Group 1. (**C**) Subtree containing isolates in Cluster BTV-22. (**D**) Subtree containing isolates clustering with BTV-22 reference sequence. Isolates from RSA are indicated based on the province they originated from (key provided in the figure).

**Figure 9 viruses-15-01611-f009:**
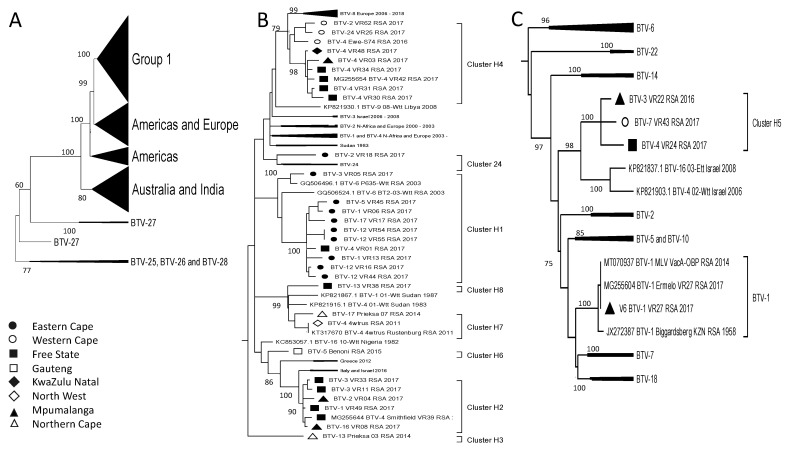
Phylogenetic analysis of Segment-9 sequences using maximum likelihood analysis. (**A**) A phylogenetic tree containing all the RSA samples grouped into a single large cluster, called Group 1. This group was divided into two subtrees represented in (**B**,**C**). (**B**) Subtree 1 from Group 1. (**C**) Subtree 2 from Group 1. Isolates from RSA are indicated based on the province they originated from (key provided in the figure).

**Figure 10 viruses-15-01611-f010:**
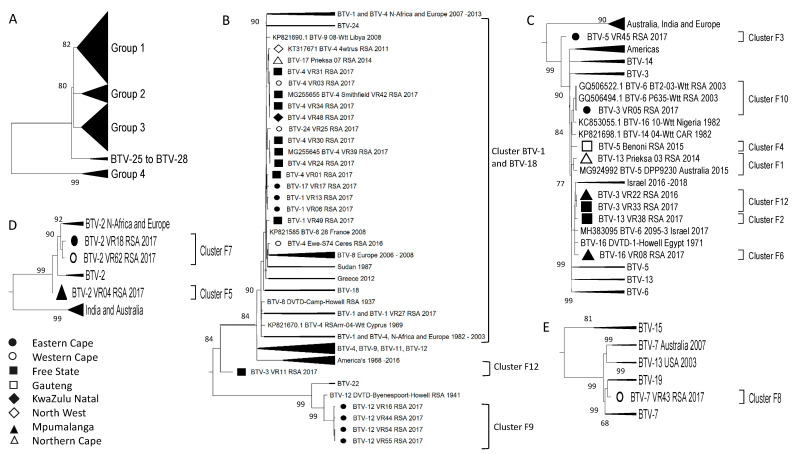
Phylogenetic analysis of Segment-7 sequences using maximum likelihood analysis. (**A**) A phylogenetic tree containing all the sequences grouped the isolates of BTV-1 to BTV-24 into four large groups. (**B**) Subtree representing Group 1. (**C**). Subtree of isolates in Group 3. (**D**) Subtree of isolates in Group 2. (**E**) Subtree of isolates in Group 4. Isolates from RSA are indicated based on the province they originated from (key provided in the figure).

**Figure 11 viruses-15-01611-f011:**
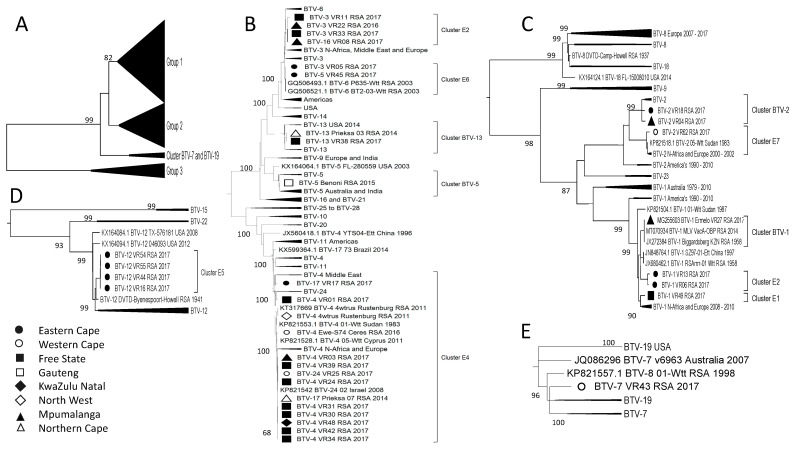
Phylogenetic analysis of Segment-6 sequences using maximum likelihood analysis. (**A**) A phylogenetic tree containing all the sequences grouping the isolates from RSA into each of the four large groups. (**B**) Subtree representing Group 1. (**C**) Subtree of isolates in Group 2. (**D**) Subtree of isolates in Group 3. (**E**) Subtrees of isolates in clusters BTV-7 and BTV-19. Isolates from RSA are indicated based on the province they originated from (key provided in the figure).

**Figure 12 viruses-15-01611-f012:**
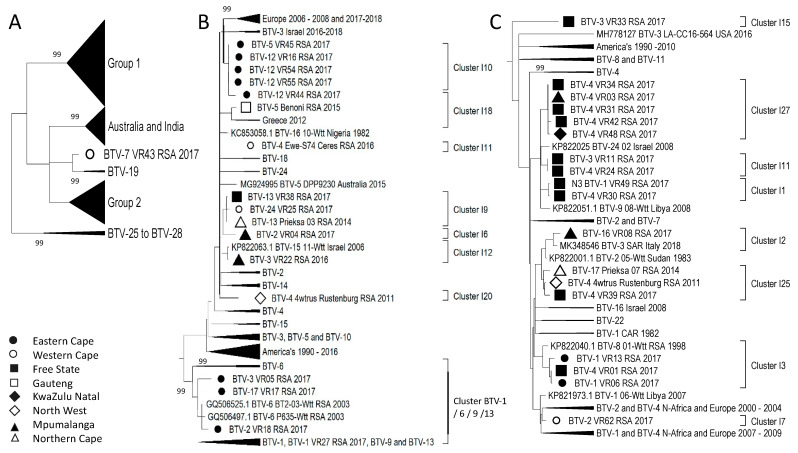
Phylogenetic analysis of Segment-10 sequences using maximum likelihood analysis. (**A**) A phylogenetic tree containing all the sequences grouping the isolates from RSA into two large groups, while sample BTV-7_VR43_RSA_2017 forms a monophyletic cluster called Cluster I8. (**B**) Subtree representing Group 1. (**C**) Subtree of isolates in Group 2. Isolates from RSA are indicated based on the province they originated from (key provided in the figure).

## Data Availability

Information on the country of origin, year of isolation, and GenBank accession number for each of the 98 genomes used in this study is provided in Appendix A.

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
