# Peer review of "Widespread Reassortment Contributes to Antigenic Shift in Bluetongue Viruses from South Africa"

_viruses, 2023, doi:10.3390/v15071611_

Round 1
Reviewer 1 Report
The authors have re-sequenced 35 BTV complete genomes from 11 different serotypes and analysed them in comparison with the available sequence information from GenBank. The work shows a comprehensive picture of genetic shift and drift. In summary, the work represents an excellent compilation and bioinformatic analysis of the current sequence information in BTV.
I would like to have one small point clarified or should be clearly formulated in the MAnuscript. The possibility of two or more virus strains in one animal or sample is not impossible. How was it ensured that only one virus was propagated and sequenced (plaque assay, limited dilution,...). If the sequenced viruses have not been processed accordingly, this should be clearly described and discussed in the paper.
Author Response
I would like to have one small point clarified or should be clearly formulated in the MAnuscript. The possibility of two or more virus strains in one animal or sample is not impossible. How was it ensured that only one virus was propagated and sequenced (plaque assay, limited dilution,...). If the sequenced viruses have not been processed accordingly, this should be clearly described and discussed in the paper.
Answer:
Thank you for the question, this is indeed something of concern to us and thus we investigate the reads mapped to the new consensus sequences. This has been included in the materials and methods of the manuscript.
Reviewer 2 Report
Review report
A brief summary: The aim of this work is to characterize the genetic profile of bluetongue virus (BTV) isolates (collected during a specific period of time) in South Africa.
Broad comments: The article provides relevant results about the genetic profiles of some South African isolates of BTV and gives insights into the molecular evolution of BTV in the field.
Specific comments
In the results section (pag 5) the authors repeat some information already provided in the materials and methods section. For example the first sentence could be removed (“Blood samples…….in South Africa”). But also the long part going from “The complete genome sequence……threshold of 5% per cluster” could be removed because the same description was already provided in the materials and methods.
Figure 2: in the histogram corresponding to segment 8, cluster 18 should be colored in green.
Author Response
In the results section (pag 5) the authors repeat some information already provided in the materials and methods section. For example the first sentence could be removed (“Blood samples…….in South Africa”). But also the long part going from “The complete genome sequence……threshold of 5% per cluster” could be removed because the same description was already provided in the materials and methods.
Answer:
Thank you kindly for the comment, the manuscript has been edited as suggested.
Figure 2: in the histogram corresponding to segment 8, cluster 18 should be colored in green.
Answer:
This picture has been corrected.
Reviewer 3 Report
This manuscript reports on the diversity of BTV isolates from RSA and compares them with vaccine strains and GenBank deposits.
One of the limitations of the study is the overwhelming preponderance of 2017 isolates in the virus isolates used - by my count 28 out of the 35 studied = 80%. That limitation should be discussed, Figure 1 also suggests that they were mostly recruited from the eastern part of the country. 2017 is also getting a bit old, were there no isolates that were younger?
There are no line numbers in the submitted manuscript, which makes reviewing and reporting a bit awkward - hopefully you will find the relevant part I am referring to:
INTRO:
The beginning of the 5th paragraph - about the general features of the virus could arguably be moved up to lead the second paragraph. The sentence from "Identification ..." could be the lead-in into the bit about the molecular epidemiology and strain diversity?
The last paragraph of the INTRO introduces elements of the M&Ms, after the first sentence, so you should move these parts into the appropriate section.
RESULTS:
The first paragraph repeats what is essentially information from the M&Ms and should remain there - delete here (apart from reference to Figures 1 and 2).
DISCUSSION
Should discuss the limitations in terms of time period more (as stated above, 80% of the isolates are representative of one year only), as well as possibly the geographical limitation.
Author Response
One of the limitations of the study is the overwhelming preponderance of 2017 isolates in the virus isolates used - by my count 28 out of the 35 studied = 80%. That limitation should be discussed, Figure 1 also suggests that they were mostly recruited from the eastern part of the country. 2017 is also getting a bit old, were there no isolates that were younger?
Answer:
As mentioned in the manuscript, BT is a neglected disease in South Africa due to the endemic nature of it. This implies that outbreaks are not reported regularly, neither are samples submitted for laboratory confirmation. This unfortunate limitation has been described in both the introduction and discussion.
There are no line numbers in the submitted manuscript, which makes reviewing and reporting a bit awkward - hopefully you will find the relevant part I am referring to:
Answer:
Our sincerest apologies for this oversight.
INTRO:
The beginning of the 5th paragraph - about the general features of the virus could arguably be moved up to lead the second paragraph. The sentence from "Identification ..." could be the lead-in into the bit about the molecular epidemiology and strain diversity?
Answer:
Changes were made as suggested
The last paragraph of the INTRO introduces elements of the M&Ms, after the first sentence, so you should move these parts into the appropriate section.
Answer:
This section has been moved to the M&M
RESULTS:
The first paragraph repeats what is essentially information from the M&Ms and should remain there - delete here (apart from reference to Figures 1 and 2).
Answer:
Changes were made as suggested
DISCUSSION
Should discuss the limitations in terms of time period more (as stated above, 80% of the isolates are representative of one year only), as well as possibly the geographical limitation.
Answer:
This concern was re-iterated in the discussion as suggested.

Round 2
Reviewer 3 Report
Thank you for the prompt revision - my suggestions have been addressed.